# A catechol-O-methyltransferase genetic variant impacts functional movement in tactical athletes

**Marcus K. Taylor**[1], **Lisa M. Hernández**[1,2]*, **Richard C. Allsopp**[3], **John J. Fraser**[1]

**1** Naval Health Research Center, San Diego, California, United States of America, **2** Leidos, Incorporated, San Diego, California, United States of America, **3** Yanagimachi Institute for Biogenesis Research, John A. Burns School of Medicine, University of Hawaii, Honolulu, Hawaii, United States of America

* lisa.m.hernandez75.ctr@health.mil

**Data Availability Statement:** The datasets generated and/or analyzed during the current study are not publicly available since public deposition would breach compliance with the protocol approved by the Naval Health Research Center

## Abstract

Functional movement is a valuable indicator of physical performance, injury risk, and/or musculoskeletal impairment following injury. However, genetic variation and gene–environment interactions that may affect functional movement are largely unexplored. We recently reported a linkage between trauma exposure and functional movement in male tactical athletes. Here, we examined the effect of a common genetic variant, rs737865, within the catechol-O-methyltransferase gene on functional movement in specialized military personnel (N = 134). We also explored whether rs737865 modulated the influence of trauma exposure. Genotyping was determined from saliva, trauma exposure was self-reported using the Brief Trauma Questionnaire, and functional movement was evaluated using the Functional Movement Screen™. The effect of rs737865 on functional movement was evaluated using general linear models, while associations between trauma and functional movement were determined with regression models. An alpha level of 0.05 was set as the threshold for significance. In the standard three-genotype model (*GG*, *AG*, *AA*), rs737865 predicted functional movement (p = 0.03, $\eta^2_p$ = 0.05). Specifically, *GG* (n = 9) exhibited the highest functional movement scores (mean [M] ± standard deviation [SD] = 17.2±1.9), followed by *AG* (n = 45; M±SD = 15.9±2.5), and then *AA* (n = 80; M±SD = 15.2±2.3). The connection between rs737865 and functional movement was preserved in both the dominant *G* model (*G* vs *AA*; p = 0.03, $\eta^2_p$ = 0.04) and the dominant *A* model (*A* vs *GG*; p = 0.03, $\eta^2_p$ = 0.03). The rs737865 variant also modulated the influence of trauma on functional movement. To our knowledge, this is the first discovery linking rs737865 to functional movement, which may lead to greater precision in musculoskeletal injury risk stratification and increased efficacy of strength and conditioning programs in tactical athletes. In resource constrained settings, genomic modeling may help to direct limited assets to at-risk subgroups (for screening purposes). It may also help to individualize, and enhance, strength and conditioning programs based on the genomic signature of a person's training response.

(NHRC) Institutional Review Board (IRB). Data may be made available after a data sharing agreement has been established. Such inquiries can be directed to the Institutional Point of Contact: NHRC IRB Office. Email: USN.NHRC.IRB@health.mil. Full citation of where data can be found: Human Subjects Protocol Number NHRC.2023.0005, The Readiness, Resilience, and Recovery (R3) Study.

**Funding:** Office of Naval Research Summer Faculty Research Program and the Defense Health Agency under work unit No. N1522.

**Competing interests:** The authors have declared that no competing interests exist.

## Introduction

Functional movement, a widely used indicator of physical performance, is linked to musculo-skeletal injury [1]. Accordingly, functional movement (FM) assessments are used in strength and conditioning settings as well as in research to characterize complex motor patterns as a correlate of injury [2]. We recently reported a connection between psychological trauma exposure to FM in tactical athletes [3]. In that study, trauma exposure was associated with lower FM, independent of age and physical injury history. This formed a framework for a mind–body connection underlying FM in tactical athletes [3]. However, aside from some demographic features (i.e., gender, age), individual differences affecting FM are largely unexplored. Advancements in this area are needed to increase the effectiveness of exercise programming and precision in clinical practice [4, 5], which will lead to more individualized plans for injury prevention, mitigation, and rehabilitation care. In resource constrained settings, genomic modeling may help to direct limited assets to at-risk subgroups (for screening purposes). It may also help to individualize, and enhance, strength and conditioning programs based on the genomic signature of a person's training response.

Genetic variation is a stable individual difference that could modulate the trauma–FM dyad. For instance, the catechol-O-methyltransferase (COMT) gene (chromosome 22q11.2) governs translation of COMT [6]. This enzyme regulates central and peripheral dopamine transmission, with downstream effects during motor planning, execution of gait [7, 8], and in other complex movements [9]. It is therefore considered a "genotypic indicator" of dopamine signaling [7]. COMT also regulates epinephrine and norepinephrine transmission, which may partially explain its observed relationships with pain [10, 11] and mental health [12, 13]. The most studied single nucleotide polymorphism of the COMT gene is rs4680, a G [valine] to A [methionine] substitution at codon 158, exon 4.25 [8, 10, 14]. Radojević et al. [8] observed a higher frequency of "freezing of gate" in Parkinson's patients who were rs4680 AA carriers in comparison to G carriers, while Yu et al. observed that Parkinson's patients with rs4680 A-carriage presented with more severe bradykinesia in the upper extremities as well as greater tremor [14]. Similarly, an 8-year longitudinal study of older adults revealed an accelerated decline in rapid gait among AA carriers, as compared with G-carriers [7]. Combined, these studies demonstrate consistent, functional implications of rs4680. Ironically, A-carriage (valine to methionine substitution) corresponds to lower COMT concentrations, resulting in lower dopamine metabolism, and consequently, higher dopamine signaling [6, 15]. This is reconciled by the tonic-phase dopamine hypothesis, which purports that high background concentrations of dopamine raise the threshold for a rapid rise and fall (i.e., phasic release) in response to transient challenge [8]. Other routinely studied COMT variants include rs737865 (an A/G variant at intron 1 of the 5' region), and rs165599 (an A/G variant located in the 3' untranslated region). Li et al. [16] identified that rs737865 (A/G) confers protection from chemotherapy-related memory decline in breast cancer patients. Specifically, A/G carriers had better memory outcomes as compared with their A/A counterparts. These findings reinforce a codominant model in which both variants are expressed, as proposed by Nogueira et al. [17]. Likewise, rs165599 was also associated with memory performance in a similar population, with A carriers having greater impairment than homozygous G carriers [18]. Although rs737865 and rs165599 appear to mediate dopamine signaling, they are surprisingly understudied with respect to physical function. It is also important to note that collectively, rs4680, rs737865, and rs165599 compose a three marker haplotype, which may be inherited together and could therefore act synergistically [19].

Whereas some studies present logical pathways between the COMT gene and physical function in older adults [8, 10, 14], others tie COMT to bodily pain [10], mental health, and

cognitive function [16]. In contrast, relatively few studies have examined the influence of the *COMT* gene on physical performance in younger, healthy, athletic populations. Tartar et al. observed a greater frequency of rs4680 *GG* in mixed martial arts participants relative to athlete, and non-athlete, comparison groups [20]. Also, Abe et al. revealed superior performance in competitive swimmers with the rs4680 *A* allele as compared with *GG* carriers [21]. By contrast, Zmijewski et al. reported no association of rs4680 with elite swimming status [22]. In an exercise intervention study, *GG* participants had a greater extent of improved executive control as compared with *A* carriers [23]. To our knowledge, there is a lack of research linking any *COMT* variant to FM in healthy, athletic populations.

To address this knowledge gap, we evaluated the relationship between *COMT* rs737865 and FM. As a secondary aim, we explored whether rs737865 moderated the influence of trauma exposure on FM. We hypothesized that rs737865 would directly associate with FM, and that it would modulate the influence of trauma exposure.

## Methods

### Experimental approach

In this laboratory-based, cross-sectional study, the primary independent variable was *COMT* rs737865, and the dependent variable was FM. Genotyping and FM tests were conducted in a sample of specialized, active duty military personnel to determine the relationships between these variables. Study participants also self-reported their trauma history so that we could determine if rs737865 moderated the influence of trauma exposure on FM.

### Participants

Participants (N = 134) were enrolled in the Explosive Ordnance Disposal (EOD) Operational Health Surveillance System, a study of U.S. Navy EOD operators. U.S. Navy EOD personnel are highly trained operators who render safe all types of explosives and specialize in diving and parachuting. The local Institutional Review Board approved this research protocol (NHRC.2015.0013) and the informed consent procedures for this study. The recruitment period began on 11 May 2015 and ended on 01 July 2020. All participants provided written informed consent, which was witnessed, and co-signed by, an Ombudsman, and the individual who administered the informed consent procedures. EOD operators between 18–50 years of age, and who were assigned to EOD Group One, were eligible for participation. There were no imposed exclusion criteria for the present evaluation.

### Procedures

**Genotyping of *COMT* rs737865.** A saliva sample was collected from each participant using the passive drool method [24]. Ten minutes prior to saliva collection, participants rinsed their mouths with water. Samples were immediately placed in a −80˚C freezer until shipped for analysis. Genotyping was performed by Salimetrics, LLC (Carlsbad, CA) in small batches using polymerase chain reaction amplification.

**Brief Trauma Questionnaire.** The Brief Trauma Questionnaire (BTQ) was used to assess historical trauma exposure. It consists of 10 yes/no questions regarding exposure to various types of potentially traumatic events (e.g., served in a war zone, been in a serious accident, experienced a natural disaster). Participants who responded to any of the questions in the affirmative were then presented with two additional questions: "Did you fear for your life?" and "Were you seriously injured physically?" The primary 10 questions were used to calculate the

sum score of trauma exposure (yes = 1, no = 0) with a range of 0 to 10 [25]. The BTQ is considered a reliable, valid measure, which parallels clinical interviews of trauma exposure [26].

**Functional Movement Screen.** The Functional Movement Screen™ (FMS; Functional Movement Systems, Chatham, VA) is a test battery comprised of fundamental movements, which require flexibility, mobility, and stability [27, 28]. This method was selected based on its generalizability to the performance of EOD operators, and for its known inter-rater and test-retest reliability. The FMS has seven individual movement patterns: deep squat, hurdle step, in-line lunge, shoulder mobility, active straight-leg raise, trunk stability push-up, and rotary stability. An ordinal scale ranging from 0 to 3 is used to score each movement; 0 (pain with any part of the movement), 1 (inability to complete the movement as instructed), 2 (movement with some compensation, but without pain), and 3 (correct movement without pain). Total FMS scores range from 0 to 21. The FMS was conducted and scored by one of three research assistants, who were cross-trained, and blinded to trauma exposure scores. The test demonstrates good to excellent interrater reliability of both novice and expert raters for all components [29]. FMS test-retest reliability has been reported with an intraclass correlation coefficient (ICC) range of 0.81–0.91 [30].

**Statistical analyses.** Data were analyzed using SPSS (Version 29.0; IBM Corp., Armonk, NY, USA). If data were missing, there were no imputations. We conducted descriptive analyses to summarize participant characteristics. All continuous variables were evaluated for normality per standard criteria (skewness between −1 and 1; kurtosis between −3 and 3) [31]. Data that did not meet these criteria were normalized to fulfill statistical modeling assumptions. A test for departure from Hardy-Weinberg equilibrium (data quality check for genomic studies) was performed via $\chi 2$ test for goodness of fit [32]. In the primary hypothesis test, the direct effect of rs737865 on FM was evaluated with general linear models. To test the second hypothesis, we first confirmed the overall, unadjusted association between trauma exposure and FM with a linear regression model. Then, we explored whether rs737865 moderated the influence of trauma exposure on FM by comparing the effect sizes that were derived from separate models, conducted at each level of the moderator (e.g., rs737865 *G* vs *AA*) [33]. Post-hoc power was determined for each model using SPSS. An alpha level of 0.05 was set as the threshold for significance.

For all models, theoretically relevant variables were screened as candidate covariates following standardized criteria. Specifically, a variable was selected as a covariate if it related (p <0.05) to the independent and dependent variable, thus qualifying it as a potential confounder or mediator [34]. Then, candidate mediators were further tested following Baron and Kenny's four-step causal steps approach [33]. Collinearity diagnostics were employed for all combined regression models using variance inflation factors (cut point = 5) [35].

## Results

Participant characteristics, *COMT* rs737865 genotype frequencies, and the FMS average total score and range for this sample can be found in Table 1. Genotype frequencies in this study did not deviate from Hardy-Weinberg equilibrium: ($\chi^2$(2, N = 134) = 0.45, p = 0.80).

### *COMT* rs737865 and functional movement

**Direct, unadjusted effects.** In the standard, three-genotype model (*GG* vs *AG* vs *AA*), rs737865 predicted FM (F(2,131) = 3.7, p = 0.03, $\eta^2_p$ = 0.05, 1−β = 0.67). FMS total scores are reported as the group mean±standard deviation. Specifically, *GG* exhibited the highest FMS scores (17.2±1.9), followed by *AG* (15.9±2.5), and then *AA* (15.2±2.3). The link between rs737865 and FM persisted in a dominant *G* model (*G* >*AA*; F(1,132) = 5.0, p = 0.03, $\eta^2_p$ =

**Table 1. Participant characteristics.**

| | n | Percent | M ± SD | Range |
|---|---|---|---|---|
| Gender | | | | |
| Male | 128 | 95.6% | | |
| Female | 5 | 3.7% | | |
| Missing | 1 | 0.07% | | |
| Race and ethnicity | | | | |
| American Indian[a] | 0 | 0.0% | | |
| Asian/Pacific Islander[a] | 0 | 0.0% | | |
| Black | 2 | 1.5% | | |
| Hispanic | 3 | 2.2% | | |
| White | 110 | 82.1% | | |
| Multiracial and/or Multiethnic | 17 | 12.7% | | |
| Missing | 2 | 1.5% | | |
| Rank | | | | |
| Enlisted | 88 | 65.7% | | |
| Officer | 45 | 33.6% | | |
| Missing / N/A | 1 | 0.7% | | |
| Age (years) | | | 32.9±6.6 | 22–50 |
| Missing | 0 | 0.0% | | |
| Military experience (years) | | | | |
| 1–2 | 1 | 0.7% | | |
| 3–5 | 37 | 27.6% | | |
| 6–9 | 25 | 18.7% | | |
| 10+ | 70 | 52.2% | | |
| Missing | 1 | 0.7% | | |
| Education | | | | |
| High school graduate/GED | 14 | 10.5% | | |
| Some college | 30 | 22.4% | | |
| Associate degree | 17 | 12.7% | | |
| Bachelor's degree or higher | 72 | 53.7% | | |
| Missing | 1 | 0.7% | | |
| Deployments | | | 2.6±1.3 | 0–4+ |
| 0–1 | 40 | 29.9% | | |
| 2–3 | 41 | 30.6% | | |
| 4+ | 52 | 38.8% | | |
| Missing | 1 | 0.7% | | |
| *COMT* rs737865 | | | | |
| *AA* | 80 | 59.7% | | |
| *AG* or *GA* | 45 | 33.6% | | |
| *GG* | 9 | 6.7% | | |
| Missing | 0 | 0.0% | | |
| Brief Trauma Questionnaire | | | 2.7±1.9 | 0–9 |
| Missing | 6 | 4.5% | | |
| Functional Movement Screen (total score) | | | 15.6±2.4 | 7–21 |
| Missing | 0 | 0.0% | | |

[a]Six participants identified as American Indian and one identified as Asian/Pacific Islander, but they also selected another race or ethnicity.

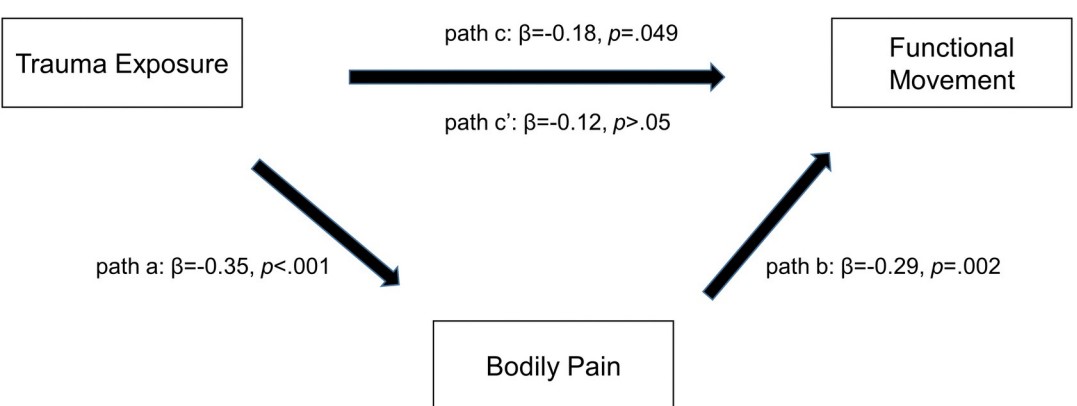

**Fig 1. Bodily pain as a potential mediator of the trauma exposure-FM relationship in all genotypes.** Candidate covariates: age, physical injury, and bodily pain. FM = functional movement.

0.04, $1-\beta = 0.60$), and in a dominant $A$ model ($A < GG$; F(1,132) = 4.7, p = 0.03, $\eta^2_p = 0.03$, $1-\beta = 0.57$). None of the candidate covariates (i.e., age, physical injury [potential confounders], or bodily pain [potential mediator]) met criteria for inclusion in any of the models.

## Trauma exposure and functional movement

**Direct, unadjusted effects.** In an unadjusted model, higher trauma exposure predicted lower FM scores (F(1,126) = 8.2, $\beta = -0.25$, p = 0.005). Age met criteria as a potential confounder, and bodily pain met criteria for further testing as a potential mediator.

**Direct, adjusted effect.** In an age-adjusted model, trauma exposure maintained a unique influence on FM (overall model F(2,125) = 7.6, p = 0.001; unique effect $\beta = -0.18$, p = 0.049).

**Mediating effect of bodily pain.** Bodily pain was evaluated as a potential mediator of the trauma exposure–FM relationship via the causal steps approach (Fig 1). The first step (path c) was established per the above. Second, age-adjusted trauma exposure was positively associated with bodily pain (path a; overall model F(2,116) = 10.5, p <0.001). Bodily pain was positively linked to FM in a model that was adjusted for both age and trauma exposure (path b; overall model F(3,115) = 8.5, p <0.001). Inclusion of bodily pain in the model reduced the original (age adjusted) effect of trauma exposure by 33.3% (path c'), implying partial mediation.

**Moderating effect of rs737865.** The direct, age-adjusted effect of trauma exposure on FM (as described above) intensified in a model of only $AA$ carriers (overall model F(2,71) = 5.9, p = 0.004), and the original effect size increased by 61.1% (Fig 2). However, in this combined model, bodily pain did not meet criteria as a mediator. Specifically, pain and trauma exposure contributed similarly. In $G$-carriers, a direct, age-adjusted effect of trauma exposure on functional movement was not observed and thus, indirect effects were not evaluated. Due to the prohibitively small subsample size of the $GG$ group, moderated effects were not explored in the three-genotype, or the dominant $A$, model.

## Discussion

To our knowledge, this is the first study to establish a link between the *COMT* rs737865 genetic variant and FM in tactical athletes. This study further confirmed that trauma exposure directly influenced FM, with rs737865 moderating the relationship. These findings have implications for injury risk screening and for assessing impairment following injury.

*COMT* rs737865 directly impacted FM, with $G$ carriers demonstrating superior FMS scores relative to $A$ carriers. This overall pattern persisted across the three-genotype, $A$-dominant,

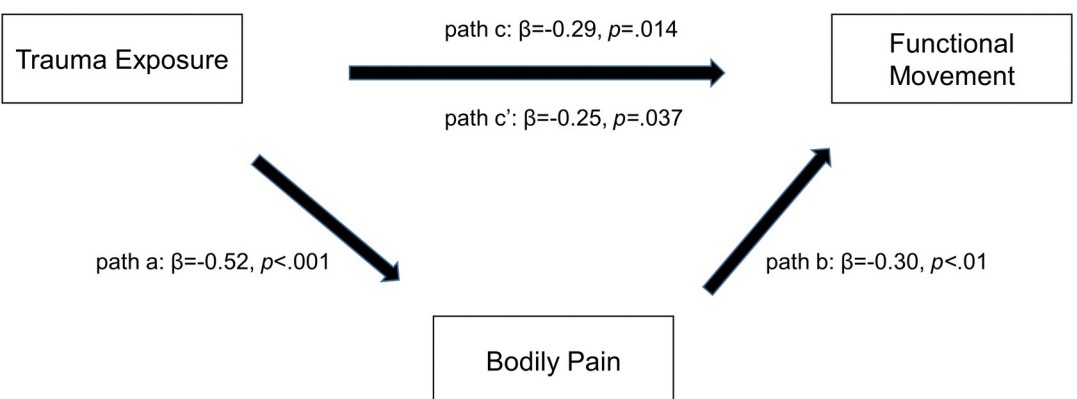

**Fig 2. Bodily pain as a potential mediator of the trauma exposure-FM relationship in *AA* carriers.** Candidate covariates: age, physical injury, and bodily pain. FM = functional movement.

and *G*-dominant models. Although previous research in clinical populations links the *COMT* gene with gait [7, 8] and other aspects of human movement [9], this is, to our knowledge, the first discovery to connect rs737865 and FM in any population. Despite the apparent functionality of rs737865 for motor control, the underlying mechanisms remain unclear. The rs737865 SNP resides on intron 1 of the *COMT* gene and is therefore not predicted to affect its structure or function; rather, it likely impacts gene expression. However, it is unknown how the *A* allele of rs737865 alone affects *COMT* production or enzymatic activity. Further, it is undetermined how resultant tonic dopamine levels from this genetic expression may govern phasic responses. This is a promising area of future research, which will have implications for precision medicine and rehabilitation practice. Since an athlete's genetic predisposition likely influences their response to rehabilitation, characterizing these individual differences could inform more targeted therapeutic modalities.

We also found that trauma exposure directly influenced FM, which is consistent with our previous finding in a smaller subset (n = 82) of the current sample [3]. The current study nearly doubles the initial sample size and conveys a stable effect of trauma exposure on FM. In a subtle divergence from our preliminary study, bodily pain displayed partial mediation of the relationship. It is plausible that trauma exposure may not only "act" upon FM through bodily pain, but also through other pathways. Replication studies in diverse populations will clarify the mediating role of bodily pain.

Until now, individual differences governing the trauma–FM relationship were unreported. Here, we showed that rs737865 moderated the influence of trauma exposure on FM, as the correlation was intensified in *AA* carriers, but disappeared in *G* carriers. This is a prototypical example of a gene–environment interaction which signifies the meaningful effects of individual differences on physical function. Mechanistically, this result also implies that effective dopamine regulation is a resilience factor, which can buffer the effects of trauma exposure. Precisely how rs737865 affects FM, remains unknown. A clearer understanding of this process could enable more biologically based hypothesis tests, and ultimately, refine individualized patient care.

There are strength/conditioning and clinical implications from this study. For primary, secondary, and tertiary prevention, genetic determination of key polymorphisms may be helpful in risk assessment for injury or reinjury. This could allow for greater precision when developing preventative training programs (e.g., gait training). Additionally, population-level screening of genetic markers in national serum repository samples may provide greater fidelity of

those markers as they relate to injury risk. Following injury, genetic disposition may be an important corollary during injury assessment/diagnosis, which would better inform interventions and the plan of care. While these suppositions warrant future investigation, there is great promise for some genetic biomarkers in the field of sports genomics.

There are some limitations of this study. Notably, a modest sample size yielded statistical power that registered below conventional standards (e.g., 0.80) [36]; however, this is counterbalanced by the observed p-value of our primary hypothesis test (p = 0.03). This translates to a 3% likelihood of achieving a Type I error, which is the relevant error of concern in the case of null rejection (versus a Type II error). Also, there is a need to solidify the mediating role of bodily pain and other contributing factors. To address these limitations, large replication studies are currently underway in our lab, with the intent to generalize to more diverse populations Nevertheless, this novel study forms a guiding model of human functional movement by an interplay of genetic and environmental forces.

## Acknowledgments

We wish to express our gratitude to all U.S. Navy EOD operators for their countless sacrifices while in service of their country. Dr. Anu Venkatesh served as a scientific advisor and provided expert review. Ms. Michelle Stoia provided editorial expertise.

MKT is an employee of the U.S. Government. This work was prepared as part of his official duties. Title 17, U.S.C. §105 provides that copyright protection under this title is not available for any work of the U.S. Government. Title 17, U.S.C. §101 defines a U.S. Government work as work prepared by a military service member or employee of the U.S. Government as part of that person's official duties. Report No. 23–81. The views expressed in this article are those of the authors and do not necessarily reflect the official policy or position of the Department of the Navy, Department of Defense, nor the U.S. Government. The study protocol was approved by the Naval Health Research Center Institutional Review Board in compliance with all applicable federal regulations governing the protection of human subjects. Research data were derived from approved Naval Health Research Center Institutional Review Board protocol number NHRC.2015.0013.

## Author Contributions

**Conceptualization:** Marcus K. Taylor.

**Data curation:** Lisa M. Hernández.

**Formal analysis:** Marcus K. Taylor.

**Funding acquisition:** Marcus K. Taylor, Lisa M. Hernández.

**Investigation:** Marcus K. Taylor, Lisa M. Hernández.

**Methodology:** Marcus K. Taylor, Lisa M. Hernández.

**Project administration:** Lisa M. Hernández.

**Resources:** Marcus K. Taylor.

**Supervision:** Marcus K. Taylor.

**Validation:** Lisa M. Hernández.

**Visualization:** Marcus K. Taylor.

**Writing – original draft:** Marcus K. Taylor.

**Writing – review & editing:** Marcus K. Taylor, Lisa M. Hernández, Richard C. Allsopp, John J. Fraser.

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
