## [Decision Letter · Decision Letter 0]

13 Nov 2024

PONE-D-24-37733A catechol-O-methyltransferase genetic variant impacts functional movement in tactical athletesPLOS ONE

Dear Dr. Hernández,

Thank you for submitting your manuscript to PLOS ONE. After careful consideration, we feel that it has merit but does not fully meet PLOS ONE’s publication criteria as it currently stands. Therefore, we invite you to submit a revised version of the manuscript that addresses the points raised during the review process.

We look forward to receiving your revised manuscript.

Kind regards,

Yearul Kabir, Ph.D

Academic Editor

PLOS ONE

Journal Requirements: When submitting your revision, we need you to address these additional requirements. 1. Please ensure that your manuscript meets PLOS ONE's style requirements, including those for file naming. The PLOS ONE style templates can be found at https://journals.plos.org/plosone/s/file?id=wjVg/PLOSOne_formatting_sample_main_body.pdf and https://journals.plos.org/plosone/s/file?id=ba62/PLOSOne_formatting_sample_title_authors_affiliations.pdf 2. Thank you for stating the following financial disclosure: "Office of Naval Research Summer Faculty Research Program and the Defense Health Agency under work unit No. N1522" Please state what role the funders took in the study.  If the funders had no role, please state: ""The funders had no role in study design, data collection and analysis, decision to publish, or preparation of the manuscript."" If this statement is not correct you must amend it as needed. Please include this amended Role of Funder statement in your cover letter; we will change the online submission form on your behalf. 3. In the online submission form, you indicated that "Data are confidential property of the U.S. Government and can only be shared under a data sharing agreement. Data sharing agreement inquiries can be sent to the corresponding author." All PLOS journals now require all data underlying the findings described in their manuscript to be freely available to other researchers, either 1. In a public repository, 2. Within the manuscript itself, or 3. Uploaded as supplementary information.This policy applies to all data except where public deposition would breach compliance with the protocol approved by your research ethics board. If your data cannot be made publicly available for ethical or legal reasons (e.g., public availability would compromise patient privacy), please explain your reasons on resubmission and your exemption request will be escalated for approval. 4. Please review your reference list to ensure that it is complete and correct. If you have cited papers that have been retracted, please include the rationale for doing so in the manuscript text, or remove these references and replace them with relevant current references. Any changes to the reference list should be mentioned in the rebuttal letter that accompanies your revised manuscript. If you need to cite a retracted article, indicate the article’s retracted status in the References list and also include a citation and full reference for the retraction notice.

**Additional Editor Comments:**

The research examined the association between the COMT rs737865 variant and FM in specialized military personnel, taking into account the influence of trauma exposure. Identifying genetic susceptibility factors linked to FM may offer new insights into its underlying causes, enabling identifying individuals at higher risk for FM. The findings suggest a connection between the COMT gene and FM, although the precise mechanisms remain unclear.

Revisions needed:

1. Enhance the abstract to convey the clinical significance of the findings more clearly, particularly in relation to injury risk assessment and rehabilitation programs for tactical athletes.

2. Expand the introduction with a more detailed discussion of the potential biological impacts of genetic variants, like rs737865, on FM beyond their role in dopamine signaling.

3. In the introduction, emphasize the practical applications of these findings in designing personalized training programs for tactical athletes.

4. Include additional detail on how these results might be generalized to other athletic or military populations.

This manuscript has the potential to be a valuable contribution to sports genomics and precision training. Minor revisions to clarify the clinical implications and provide a deeper exploration of genetic mechanisms will further enhance its impact.

Reviewers' comments:

Reviewer's Responses to Questions

**Comments to the Author**

1. Is the manuscript technically sound, and do the data support the conclusions?

Reviewer #1: Yes

Reviewer #2: Yes

2. Has the statistical analysis been performed appropriately and rigorously? 

Reviewer #1: Yes

Reviewer #2: Yes

3. Have the authors made all data underlying the findings in their manuscript fully available?

Reviewer #1: No

Reviewer #2: No

4. Is the manuscript presented in an intelligible fashion and written in standard English?

Reviewer #1: Yes

Reviewer #2: Yes

5. Review Comments to the Author

Reviewer #1: Based on my assessment of the manuscript:

1. Technical Soundness:

a) The manuscript is technically sound in terms of its experimental design and approach. The research addresses an important gap by investigating the influence of the COMT rs737865 variant on functional movement, as well as its interaction with trauma exposure, in a specialized population of military personnel.

b) The methodology is clearly described, including the use of the Functional Movement Screen (FMS), genotyping, and trauma exposure assessment. The choice of these methods is appropriate for the research objectives.

2. Data Support for Conclusions:

a) The data presented support the conclusions drawn by the authors. There is a clear correlation between the rs737865 variant and functional movement, as well as a significant interaction between trauma exposure and the genotype. The findings related to gene-environment interactions (involving the rs737865 variant and trauma) are compelling and consistent with the data.

b) The conclusions about how rs737865 could be used in injury risk stratification and strength and conditioning programs are reasonable based on the findings.

3. Statistical Analysis:

a) The statistical analysis appears to have been performed appropriately. The authors used general linear models to assess the effects of rs737865 on functional movement, as well as regression models for trauma exposure. Covariates were screened and tested in line with established statistical approaches.

b) The reported p-values and effect sizes (η2p) are within acceptable ranges, and the alpha threshold (0.05) is standard. However, the post-hoc power analysis indicates that the sample size might have been modest, which the authors acknowledge as a limitation. Although the power falls slightly below conventional standards, the primary findings are statistically significant (p = 0.03), which mitigates concerns of Type I error.

4. Data Availability:

a) The manuscript states that the data underlying the findings are confidential and part of U.S. Government property. This limits the public availability of data, which may be an issue depending on journal policies. However, the authors provide a contact method for data-sharing requests, which adheres to ethical requirements considering the sensitive nature of the sample (military personnel).

5. Clarity and Language:

a) The manuscript is written in clear, standard English, and the overall presentation is intelligible. The structure follows a logical progression, with clear sections for methods, results, and discussion. The scientific language is appropriate for a specialized audience.

Reviewer #2: Title and Abstract:

1. The title is informative and specific, effectively encapsulating the primary focus on the impact of the COMT rs737865 genetic variant on functional mobility in tactical athletes.

2. The abstract offers a succinct summary; it could be improved by a more explicit expression of the clinical significance of the findings, particularly in the context of injury risk assessment and rehabilitation programs for tactical athletes.

Introduction:

1. The introduction effectively establishes the connection between functional movement (FM), injury risk, and genetic factors, with a particular emphasis on the COMT gene and previous research on the relationship between trauma and FM.

2. Additional profundity could be achieved by providing a more detailed explanation of the potential biological effects of genetic variants such as rs737865 on FM beyond dopamine signaling.

3. The potential applications of the findings in personalized training programs for tactical athletes could be more explicitly highlighted in the introduction.

Methods

1. The methodology is comprehensive, with a particular emphasis on the recruitment of participants, genetic analysis, trauma exposure assessment, and FM evaluation.

2. Although the genotyping and FM assessment procedures are explicitly defined in the study, a flow diagram that delineates the study design from participant recruitment to data analysis could be beneficial in comprehending the study's structure.

3. The Functional Movement Screen (FMS) was selected over other FM assessments, and the study could benefit from a concise explanation of its alignment with the study's objectives.

Results

1. The results are presented in a plain manner, demonstrating statistically significant associations between the rs737865 variant and FM scores.

2. The sample size is identified as a constraint, and further power calculations could verify this. To be more precise, a larger sample size could have enabled a more conclusive analysis of the rs737865 and FM interaction by enhancing statistical power.

3. Further exploration of this subgroup could provide a more comprehensive understanding of the findings, as rs737865 predominantly moderated the trauma-FM relationship in AA carriers.

Discussion

1. The findings are effectively contextualized within the broader context of genetics, FM, and trauma exposure in the discussion.

2. The study's findings on the interaction between rs737865 and trauma on FM are valuable; however, it would be beneficial to provide further detail on how these results could be generalized to other athletic or military populations.

3. A discussion of potential gene-environment interactions in relation to other COMT variants (e.g., rs4680) could add substance, as rs737865, rs4680, and rs165599 frequently co-occur as a haplotype.

4. The study could be improved by providing a more explicit statement regarding the specific impact of these insights on tactical training protocols, which would increase the practical relevance of the findings.

Overall recommendation

This manuscript offers new perspectives on the correlation between FM and the COMT rs737865 variant in tactical athletes. The results have significant implications for the development of personalized strategies for injury prevention and rehabilitation. This manuscript could be a valuable contribution to the field of sports genomics and precision training with minor revisions to improve the lucidity of the clinical implications and a more thorough examination of the genetic mechanisms involved..

6. PLOS authors have the option to publish the peer review history of their article (what does this mean?). If published, this will include your full peer review and any attached files.

Reviewer #1: No

Reviewer #2: **Yes: **Mohammad Safiqul Islam

---

## [Author Response · Author response to Decision Letter 0]

4 Dec 2024

The authors wish to thank the Editor and Reviewers for their time and feedback. Our point-by-point responses can be found in the 'Response to Reviewers' document.

---

## [Editor Report · Decision Letter 1]

6 Dec 2024

A catechol-O-methyltransferase genetic variant impacts functional movement in tactical athletes

PONE-D-24-37733R1

Dear Dr. Hernández,

We’re pleased to inform you that your manuscript has been judged scientifically suitable for publication and will be formally accepted for publication once it meets all outstanding technical requirements.

Kind regards,

Yearul Kabir, Ph.D

Academic Editor

PLOS ONE
---

## [Editor Report · Acceptance letter]

23 Jan 2025

PONE-D-24-37733R1 

PLOS ONE

Dear Dr. Hernández, 

I'm pleased to inform you that your manuscript has been deemed suitable for publication in PLOS ONE. Congratulations! Your manuscript is now being handed over to our production team.

Kind regards, 

on behalf of

Dr. Yearul Kabir 

Academic Editor

PLOS ONE